# Impact of Levosimendan and Its Metabolites on Platelet Activation Mechanisms in Patients during Antiplatelet Therapy—Pilot Study

**DOI:** 10.3390/ijms25031824

**Published:** 2024-02-02

**Authors:** Joanna Sikora, Krzysztof Pstrągowski, Aleksandra Karczmarska-Wódzka, Patrycja Wszelaki, Katarzyna Buszko, Zbigniew Włodarczyk

**Affiliations:** 1Research and Education Unit for Experimental Biotechnology, Department of Transplantology and General Surgery, Collegium Medicum, Nicolaus Copernicus University, 85-094 Bydgoszcz, Poland; akar@cm.umk.pl (A.K.-W.); wszelakipatrycja@cm.umk.pl (P.W.); 2Department of Cardiology and Internal Medicine, Antoni Jurasz University Hospital No. 1 in Bydgoszcz, 85-094 Bydgoszcz, Poland; pstragowski.krzysztof@gmail.com; 3Department of Theoretical Foundations of Biomedical Science and Medical Informatics, Collegium Medicum, Nicolaus Copernicus University, 85-094 Bydgoszcz, Poland; buszko@cm.umk.pl; 4Department of Transplantology and General Surgery, Collegium Medicum, Nicolaus Copernicus University, 85-094 Bydgoszcz, Poland; wlodarczyk@cm.umk.pl

**Keywords:** platelets, ADP, levosimendan, aggregation, antiplatelet therapy

## Abstract

Levosimendan is used for the short-term treatment of severe heart failure or other cardiac conditions. The area of existing clinical applications for levosimendan has increased significantly. This study aimed to assess whether levosimendan and its metabolites impact the mechanisms related to platelet activation. In this study, we included patients with coronary artery disease receiving antiplatelet therapy. We analyzed the pharmacodynamic profile using three independent methods to assess platelet activity. The results of the conducted studies indicate a mechanism of levosimendan that affects the function of platelets, causing higher inhibition of platelet receptors and, thus, their aggregation. It is essential to clarify whether levosimendan may affect platelets due to the need to maintain a balance between bleeding and thrombosis in patients treated with levosimendan. This is especially important in the case of perioperative bleeding. This study was conducted in vitro; the research should be continued and carried out in patients to check the complete pharmacokinetic and pharmacodynamic profile.

## 1. Introduction

Levosimendan is an inodilator indicated for the short-term treatment of severe heart failure or other cardiac situations where inotropic support is considered appropriate [1,2]. The area of existing clinical applications for levosimendan has increased significantly. Currently, it also includes advanced heart failure, cardiogenic shock, takotsubo cardiomyopathy, and cardiac surgery. In addition, right ventricular failure, pulmonary hypertension, intensive care, and emergency medicine areas are also mentioned [3,4]. Levosimendan indirectly improves peripheral organ function, resulting from the drug’s inotropic effect on cardiac function [5]. A review of the current literature shows the areas of the pharmacological effects of the drug related to the pleiotropic action of levosimendan on the patient, which extends significantly above the inotropic effect used in critical patients. The described pleiotropic effects are anti-inflammatory, antioxidant, and antiapoptotic, directly affecting organs such as the kidneys, liver, intestines, lungs, and respiratory muscles [6,7]. A screening of the current literature also indicates the potential use of levosimendan in treating multisystem disease after severe acute respiratory syndrome coronavirus 2 SARS-CoV-2 infection [8,9,10]. Levosimendan represents an effective and safe treatment option for patients afflicted with novel coronavirus infections complicated by myocardial injury and cardiac insufficiency [7,9]. Moreover, considering the potential benefits of levosimendan in improving cardiac function, future research could explore its application in managing other cardiac complications associated with viral infections to expand its therapeutic potential.

Levosimendan is an inotrope agent with a unique dual mechanism of action [11]. It is a calcium sensitizer, which interacts with the Ca^2+^-saturated troponin C (TnC) and increases cardiac contractility by enhancing the sensitivity of the myocardium to calcium. Levosimendan also has a vasodilator effect due to its potential to open adenosine triphosphate (ATP)-dependent sarcolemma K^+^ channels of vascular smooth muscle cells and myocytes. Consequently, it allows their hyperpolarization and results in smooth muscle relaxation [12,13]. Moreover, levosimendan and its metabolites, via structural similarities with phosphodiesterase (PDE) inhibitors, have the potential to increase intracellular concentrations of cyclic adenosine monophosphate (cAMP) [14,15]. Several studies have demonstrated that levosimendan may influence platelets [16,17,18,19,20,21,22]. It is important to explain whether it is possible that levosimendan or the metabolites of levosimendan have a mechanism of action that affects the platelet activation cascade.

This study aimed to assess whether levosimendan and its metabolites impact the mechanisms related to platelet activation. Due to the pharmacokinetics of levosimendan, we conducted a study in which we also checked the potential effect of metabolites on platelets. The study was a pilot study, the results of which we plan to use to design an in vivo study.

## 2. Results

### 2.1. Population Baseline Characteristics

The study was planned to include up to 20 patients with CAD, who were assigned to one of two arms. Patients received aspirin or aspirin and clopidogrel based on medical recommendations. All patients gave venous blood for the experiment. The study site was the Department of Cardiology and Internal Medicine, Antoni Jurasz University Hospital No. 1 in Bydgoszcz, Poland. The baseline characteristics are presented in the table below (Table 1). Due to the fact that the study was conducted in vitro, only blood from patients was used for the test in a test tube. Levosimendan and metabolites were administered into the blood in the test tube.

### 2.2. Pharmacodynamics

Assessment of the pharmacodynamic profile of aspirin with the addition of levosimendan or its metabolites, or of clopidogrel and aspirin with the addition of levosimendan or its metabolites, was performed for each study participant using three independent methods. According to an in vitro study, levosimendan, as well as OR-1855 and OR-1896, inhibit platelet aggregation induced by ADP. Platelet activity evaluated by Multiplate, VASP, and LTA generally showed statistically significant differences between baseline vs. levosimendan, OR-1855, OR-1896, and mix (Table 2).

The mean ± SD of individual patients of on-treatment platelet reactivity evaluated with MEA was significantly greater in patients treated with aspirin alone compared with aspirin and clopidogrel together (49.40 ± 15.54 vs. 31.90 ± 32.62) (Figure 1). The data showed reduced activity of ADP receptors for all addition blood samples, indicating more strongly inhibited platelet aggregation after the addition of levosimendan (13.90 ± 8.41 vs. 4.90 ± 5.86) or its metabolites OR-1896 (16.00 ± 9.15 vs. 5.60 ± 7.12) and OR-1855 (35.70 ±15.74 vs. 15.30 ± 12.27), and a mix of levosimendan, OR-1896, and OR-1855 (10.10 ± 5.72 vs. 3.10 ± 3.84) to patients’ blood compared to the initial sample—the base sample (means ± SD of ASA vs. means ± SD of ASA and clopidogrel).

The assessment of platelet reactivity with the VASP assay shows statistical differences in platelet inhibition between patients receiving the aspirin and those receiving the aspirin and clopidogrel (73.53 ± 32.92 vs. 47.01 ± 29.01). Similarly, antiplatelet effects were observed after the addition of levosimendan (22.36 ± 20.69 vs. 17.26 ± 17.74) or its metabolites OR-1896 (36.19 ± 37.11 vs. 8.61 ± 9.03) or OR-1855 (32.60 ± 30.61 vs. 24.89 ± 21.34), and a mixture of levosimendan and both metabolites OR-1896 and OR-1855 (23.16 ± 29.03 vs. 2.15 ± 4.68) (means ± SD of ASA vs. means ± SD of ASA and clopidogrel) (Figure 2).

The results of LTA using an assay that evaluates the pathway of platelet inhibition through receptors for ADP are similar to those of the previously presented methods. There were no such significant differences in platelet reactivity between groups when assessed by LTA. Patients receiving aspirin and clopidogrel together showed more pronounced platelet inhibition than patients taking aspirin alone in the sample after the addition of the metabolite OR-1855 (85.70 ± 70.01 vs. 10.21 ± 20.05). Still, the data after the addition of levosimendan (11.35 ± 16.47 vs. 17.32 ± 27.28) or its metabolite OR-1896 (12.49 ± 14.92 vs. 8.35 ± 7.82), and a mixture of levosimendan, OR-1896, and OR-1855 (9.42 ± 13.65 vs. 5.82 ± 6.86), still showed inhibition of platelet aggregation (means ± SD of ASA vs. means ± SD of ASA and clopidogrel) (Figure 3).

## 3. Discussion

Our study is the first to evaluate levosimendan’s influence and its metabolites on platelets. It was performed using blood from a group of patients with CAD, who are potential future levosimendan recipients. We selected two classes of antiplatelet drugs with different mechanisms of action to verify whether any of these pathways are related to the antiplatelet effect of levosimendan. In the literature, two studies indicated a possible mechanism using collagen [14,17]. Disruption of platelet release may be due to a defect in platelet signal transduction, including G protein activation, phospholipase C (PLC) activation, calcium mobilization, and tyrosine phosphorylation. Furthermore, via structural similarities with PDE inhibitors, levosimendan and its active metabolite can potentially increase intracellular concentrations of cAMP [ [16],[17],[18],[19],[20],[21],[22]]. Levosimendan triggers nitric oxide (NO) production in endothelial cells by activating specific cellular pathways involving key proteins known as p38 mitogen-activated protein kinases, extracellular signal-regulated kinase, and protein kinase [23]. NO exerts important vasodilatory, antiplatelet, antioxidant, antiadhesive, and antiproliferative effects. Although endothelium-derived NO is of prime importance in cardio- and vasculoprotection, until recently, little was known about the role of platelet-derived NO. New evidence suggests that NO synthesized by platelets regulates platelet functions, particularly suppressing platelet activation and intravascular thrombosis. Moreover, platelet NO biosynthesis may be decreased in patients with cardiovascular risk factors or with coronary heart disease, and this may contribute to arterial thrombotic disease in these patients [24].

Platelet activation is a key process in both protective hemostasis and pathological thrombosis. The mode of action of levosimendan on platelets is unknown. However, platelet activation has multiple pathways through binding of several receptors. Platelet agonists interact with specific receptors on the platelets’ surface, and this integration triggers the activation of thrombocytes. Adenosine diphosphate (ADP) affects platelets via three types of purinergic receptors: P_2×1_, P_2_Y_1,_ and P_2_Y_12_. The first is associated with a calcium channel that allows Ca^2+^ to flow into the cell and is activated by ATP; ADP is its antagonist. Two more are associated with glycoprotein VI and work together to achieve complete platelet aggregation. ADP stimulates the P_2_Y_1_ receptor to create two secondary messengers: DAG and inositol 1,4,5-triphosphate (IP_3_). DAG mediates the influx of Ca^2+^ ions. An increase in the concentration of Ca^2+^ ions in platelets activates phospholipase A_2_, which in turn releases arachidonic acid (AA) from the phospholipid membranes of thrombocytes. Cyclooxygenases convert AA into prostaglandin H2 (PGH_2_), which is an intermediate product of the transformation cycle and undergoes further transformation into the internal cyclic PGG_2_ and PGH_2_ prostaglandin peroxides, from which thromboxane A_2_ is produced in the platelets and prostacyclin in the vascular endothelium. The essential ADP-specific receptor is P_2_Y_12_. Its stimulation inhibits the action of AC, which produces cAMP molecules from ATP. cAMP is a potent inhibitor of platelet aggregation [25,26,27]. We conducted the study on blood obtained from patients with CAD due to their clinical condition, which may have different platelet properties than blood from healthy people. We selected two classes of antiplatelet drugs with different mechanisms of action to verify whether any of these pathways are related to the antiplatelet effect of levosimendan. Although platelets collected from patients were inhibited by the antiplatelet therapy used (aspirin as a thromboxane inhibitor acting on the TxA_2_ receptor and clopidogrel as an ADP receptor agonist acting on the ADP receptor), levosimendan still additionally inhibited the action of platelets. This suggests that its platelet mechanism of action is different (through other receptors or a different mechanism of action). It is essential to look for a place on the platelets’ surface that is affected by levosimendan.

Several in vitro trials indicated a potential effect of levosimendan on the platelets of healthy volunteers. This study of two new groups of patients, one of which used only ASA as an antiplatelet drug and the other had dual antiplatelet therapy (ASA and clopidogrel) is particularly important in the case of antiplatelet therapy, where additional administration of levosimendan may significantly increase the risk of bleeding. Kaptan et al. [16] also demonstrated an inhibitory effect of levosimendan on platelet aggregation induced by ADP and collagen. Twelve healthy male volunteers participated in the study, divided into three groups (different concentrations of levosimendan: 10, 25, and 45 ng/mL). Aggregation was measured by only one method—LTA. The results obtained show that levosimendan reduces aggregation induced by both ADP and collagen. The ability to inhibit aggregation was directly proportional to the concentration of levosimendan. The study results showed that levosimendan significantly inhibited ADP-induced platelet aggregation. Primary aggregation was a direct consequence of agonist stimulation, while secondary aggregation was caused by releasing the granular content inside the platelets. In this case, levosimendan could contribute to the depletion of granularity resources, which affected the impairment of secondary platelet aggregation [15]. Also, Plaschke et al. [17] studied the effect of levosimendan on platelet aggregation in vitro, obtaining similar results, thus confirming the anti-aggregatory effect of levosimendan. Three healthy volunteers were given venous blood for the experiment. Platelet agonists were, again, ADP and collagen. The results of this study showed that there was only a relationship between high levosimendan concentrations and the inhibition of platelet aggregation, which was dependent on cAMP concentration. The authors suggested that the inhibition of aggregation observed after levosimendan was due to the inhibition of PDE 3 [17].

In turn, Bent et al. conducted a similar study on an animal model. They obtained different results, which may indicate that the results obtained in vitro do not correlate with the results of in vivo tests. A total of 40 rats were used, which were randomly divided into four groups of 10 individuals each: a control group, a study group receiving levosimendan bolus, a lipopolysaccharide group, and a lipopolysaccharide study group receiving lipopolysaccharide and levosimendan bolus. Aggregation was measured using an MEA. Aggregation was induced by two agonists separately: ADP and collagen. Levosimendan administered at a clinically relevant dose in vivo did not significantly affect platelet activity as opposed to high in vitro doses [20]. A study designed by Krychtiuk et al. evaluated the effect of levosimendan on the thrombotic phenotype of human cells isolated from the human endothelium and human cells isolated from the umbilical vein. The results showed that thrombin significantly increases the activity of plasminogen activator inhibitor-1 (PAI-1) activity and tissue factor (TF) in the cells studied. At the same time, the administration of levosimendan significantly inhibited the activity of PAI-1 and TF, which confirms its antithrombotic effect. However, it is necessary to ensure these results in vivo [28].

During the metabolism of levosimendan, approximately 5% of the administrated drug is reduced in the large intestine by bacteria towards the amino phenolpyridazinone metabolite OR-1855. OR-1855 is further metabolized in the liver by acetylation (N-acetyltransferase-2) to form the active metabolite OR-1896. OR-1896 and OR-1855 are formed slowly and are detectable 12 h after starting a continuous infusion. The mean protein-binding values are 98% for levosimendan, 39% for OR-1855, and 42% for OR-1896 [20]. Levosimendan has an elimination half-life of 1–1.5 h, so concentrations of this drug decrease rapidly after stopping the infusion. The concentrations of the metabolites after stopping the infusion continue to increase and reach maximum concentrations 48–78 h after starting a 24 h infusion. The mean elimination half-life of the metabolites was 70–80 h [29]. Pataricza et al. studied the effect of levosimendan on platelet aggregation in vitro and its modulation in the presence of albumin. Platelet-rich plasma and washed platelets from nineteen healthy volunteers were used for the study. Levosimendan is highly bound to plasma proteins. Therefore, a change in albumin levels may change the effect of the drug. Ultimately, researchers confirmed that increasing albumin levels weakens the anti-aggregative effect of levosimendan, which is of clinical significance [18]. The designed study also indicates that levosimendan and its active metabolites persist in the patient’s blood much longer than the drug itself (up to 80 h after the end of the infusion) and have a platelet-inhibiting effect in vitro. It is worth noting that levosimendan is 95–98% bound to plasma proteins, which will directly impact its concentration in patients’ blood and, therefore, its therapeutic effect [30]. The degree of drug binding to plasma proteins is one of the primary factors determining the duration and strength of its action. In the case of solid binding of the drug to proteins (e.g., 98%), only about 2% of the absorbed drug dose remains in the blood as the free fraction, and only this part can penetrate the tissues and, therefore, exert a pharmacological effect. The degree of binding to plasma proteins depends on, among other factors, the presence of inflammation, kidney and liver diseases, and age. Only the free form of the compound is biologically active and can overcome biological barriers. However, in states of hypoalbuminemia, which may be observed in this group of patients, the effect of drugs bound by albumin is intensified due to the increase in the pharmacologically active fraction [31]

Confirming whether levosimendan or its metabolites have some effect on platelets is of clinical importance. This was proven by a study conducted by Lahtinen et al. [32]. In this experiment, it was analyzed whether the administration of levosimedan to patients after cardiac surgery carries the risk of increased bleeding. After heart valve surgery, two hundred patients took part in the test. The results obtained indicate that postoperative bleeding increased by 31% in the levosimendan group compared to the placebo group.

### Study Limitations

The study population was exclusively comprised of patients with CAD. Thus, its results may not fully reflect the profile of levosimendan and its metabolites in patients presenting with heart failure or cardiac surgery patients. The possibility that insufficient study participants may have influenced the study results should also be considered. Only one platelet inhibition pathway has been tested, so it is still unknown if other aggregation mechanisms are also sensitive to levosimendan. The study was conducted in vitro; the research should be continued and carried out in patients to check the complete pharmacokinetic and pharmacodynamic profile.

## 4. Materials and Methods

### 4.1. Study Design and Population

The study was designed as a single-center, in vitro pilot study conducted according to the Declaration of Helsinki and Good Clinical Practice guidelines. The protocol of the study was approved by the Ethics Committee of the Nicolaus Copernicus University in Toruń, Collegium Medicum in Bydgoszcz (approval number KB 185/2015). Each patient provided written informed consent to participate in the study before recruitment. Patients eligible for enrollment were males or non-pregnant females aged 18–80 years with a diagnosis of coronary artery disease (CAD). Key exclusion criteria included ongoing (or terminated within preceding 14 days) treatment with any P_2_Y_12_ receptor inhibitor, treatment with oral or parenteral anticoagulants, history of intracranial hemorrhage or recent (defined as last 30 days) gastrointestinal hemorrhage, and coagulation disorders at the time of screening. The complete list of inclusion/exclusion criteria is presented in Table 3.

CAD patients admitted to the Department of Cardiology, Antoni Jurasz University Hospital in Bydgoszcz, Poland, were screened for eligibility for the study, who orally received a 300 mg loading dose (LD) of aspirin or 300 mg LD of aspirin and 75 mg of clopidogrel. Each study participant was treated according to the latest ESC guidelines for managing patients presenting with CAD. After the eligibility screening and the provision of informed consent, a blood sample for evaluation was collected 2 h after the drug administration. All enrolled patients provided written informed consent to participate in the trial. The study was planned to include up to 20 patients with CAD, who were assigned to one of two arms, according to Figure 4.

### 4.2. Methods

Levosimendan and metabolites OR-1896 and OR-1855 were in the appropriate purity category (pure for analysis) and complied with Good Laboratory Practice, Good Manufacturing Practice, and Good Clinical Practice criteria. Levosimendan and metabolites OR-1896 and OR-1855 were obtained from OrionPharma (manufacturer of Simdax). Levosimendan and metabolite concentrations were calculated from the available literature for clinically relevant doses in vivo. The target concentrations were levosimendan at 45 ng/mL, and metabolite OR-1986 and metabolite OR-1855 at 4 ng/mL each.

Levosimendan and metabolites were stored at −80 °C in single portions prepared for use when the patient was included in the research study and a blood sample was taken. Reagents were added to blood collected from the patient and incubated at room temperature for 25 min (according to recommendations and my own experience with optical and impedance aggregometry). In the next step, the appropriate reagents were added, mixed several times, and incubated for 15 min at 37 °C. After this time, the measurement procedures for the methods used in the research study were started. The same procedure was followed for the control sample (to which neither levosimendan nor metabolites were added), this sample being the reference for all test samples.

Platelet reactivity was evaluated at predefined time points using three different methods: multiple-electrode aggregometry (MEA) performed with the multiplate analyzer (ADPtest, Roche Diagnostics, Switzerland), vasodilator-stimulated phosphoprotein (VASP) with the use of Flow Cytometry (BioCytex, Inc., Marseille, France), and impedance aggregometry (LTA) with the use of ADP as an agonist (Chrono-Log Corporation, Havertown, PA, USA).

### 4.3. Statistical Analysis

Data for pharmacodynamic variables were presented as means with standard deviations (SD). Data for age were given as the median. Continuous variables were compared between the study groups with Student’s *t*-test and Mann–Whitney U test, depending on the presence or absence of the normal distribution (as assessed by the Shapiro–Wilk test). The chi-square test performed comparisons between categorical variables. Statistical calculations were performed using the Statistica 13 package (StatSoft, Tulsa, OK, USA) and Matlab R2014 (Matlab and Statistics Toolbox Release 2014, The MathWorks Inc., Natick, MA, USA).

## 5. Conclusions

The results of the conducted study indicate that a mechanism of levosimendan affects the function of platelets, causing higher inhibition of platelet receptors and, thus, their aggregation. It is essential to clarify whether levosimendan may affect platelets due to the need to maintain a balance between bleeding and thrombosis in patients treated with levosimendan. This is especially important in the case of perioperative bleeding, and is a fundamental observation because the concentrations of the metabolites after stopping the infusion continue to increase and reach maximum concentrations 48–78 h after starting a 24 h infusion. Further research is warranted to put these findings into a clinical perspective.

## Figures and Tables

**Figure 1 ijms-25-01824-f001:**
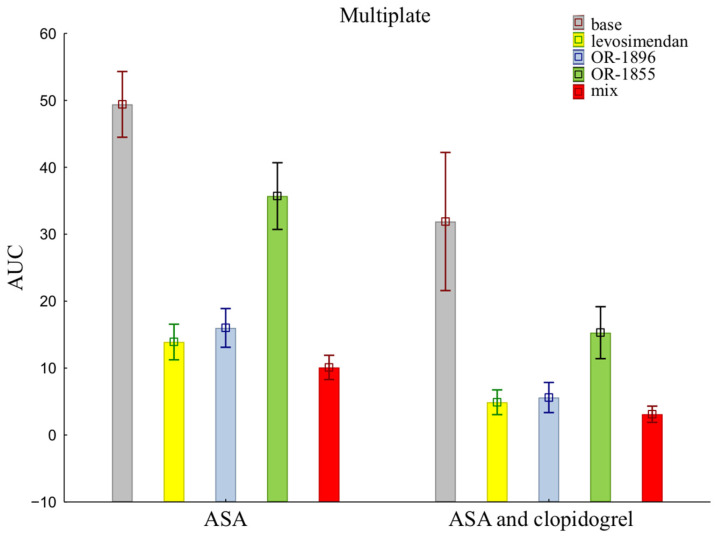
Platelet reactivity according to multiplate assay; ASA: acetylsalicylic acid; AUC: area under the curve; OR-1855: active metabolite of levosimendan; OR-1896: active metabolite of levosimendan; mix: blood with levosimendan and both active metabolites.

**Figure 2 ijms-25-01824-f002:**
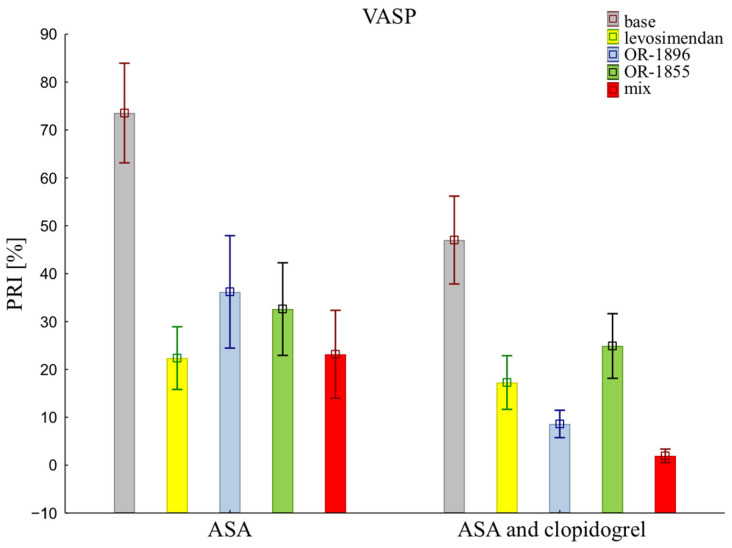
Platelet reactivity according to VASP assay; VASP: vasodilator-stimulated phosphoprotein; ASA: acetylsalicylic acid; PRI: platelet reactivity index; OR-1855: an active metabolite of levosimendan; OR-1896: an active metabolite of levosimendan; mix: blood with levosimendan and both active metabolites.

**Figure 3 ijms-25-01824-f003:**
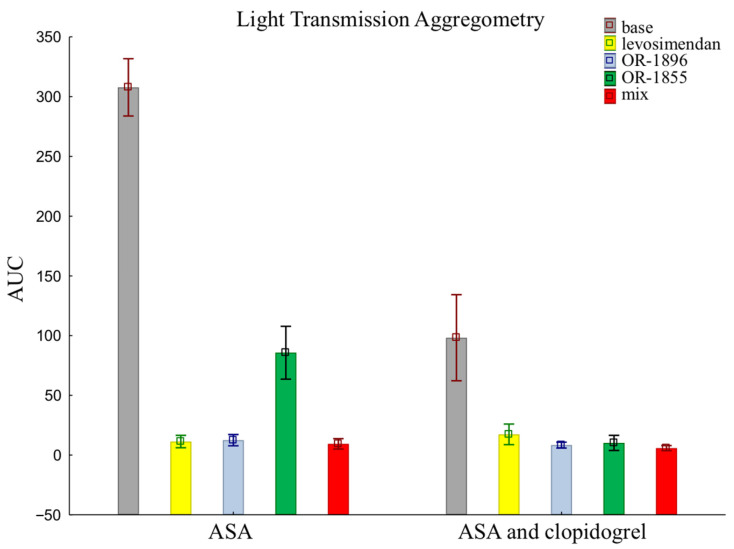
Platelet reactivity according to LTA assay; LTA: light transmission aggregometry; ASA: acetylsalicylic acid; AUC: area under the curve; OR-1855: active metabolite of levosimendan; OR-1896: active metabolite of levosimendan; mix: blood with levosimendan and both active metabolites.

**Figure 4 ijms-25-01824-f004:**
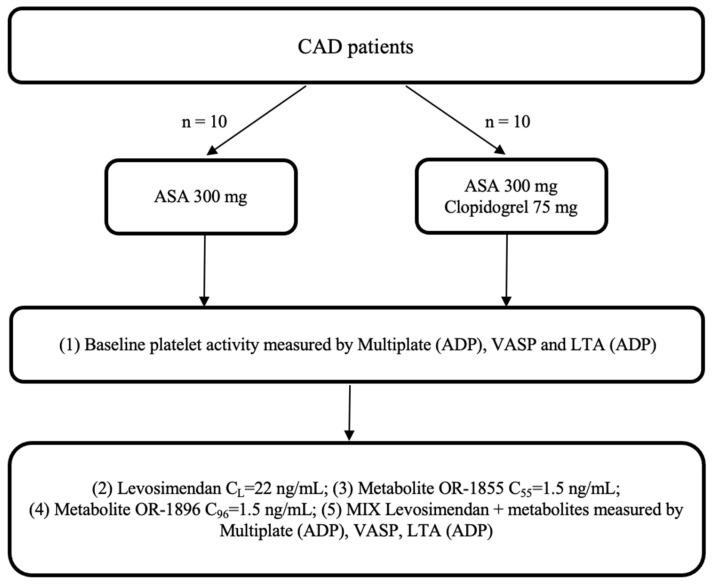
Trial flowchart: CAD: coronary artery disease; ASA: acetylsalicylic acid; ADP: adenosine diphosphate; LTA: light transmission aggregometry; VASP: vasodilator-stimulated phosphoprotein; OR-1855: active metabolite of levosimendan; OR-1896: active metabolite of levosimendan; mix: blood with levosimendan and both active metabolites.

**Table 1 ijms-25-01824-t001:** Baseline characteristics of trial participants.

	ASA	ASA and Clopidogrel	*p* Value
Age (years)	62.8 *	65.6 *	0.781
Male	40 (4)	60 (6)	0.656
Diabetes mellitus	40 (4)	10 (1)	1.000
Hypertension	60 (6)	50 (5)	1.000
Current smoker	10 (1)	60 (6)	0.057
Prior acute stroke	20 (2)	20 (2)	1.000
Hyperlipidemia	60 (6)	10 (1)	0.057

Data are shown as percentage of patients and number of patients (n), * data are shown as median; ASA: acetylsalicylic acid; AMI: acute myocardial infarction.

**Table 2 ijms-25-01824-t002:** Comparison of platelet reactivity results described as *p* values in both groups.

	Multiplate (ADP)	VASP	LTA (ADP)
ASA group (*p* values)			
base vs. levosimendan	0.006	0.013	0.006
base vs. OR-1896	0.006	0.047	0.006
base vs. OR-1855	0.054	0.006	0.006
base vs. mix	0.006	0.006	0.006
ASA and clopidogrel group (*p* values)			
base vs. levosimendan	0.006	0.007	0.075
base vs. OR-1896	0.006	0.006	0.060
base vs. OR-1855	0.007	0.060	0.016
base vs. mix	0.006	0.007	0.037

*p* values were calculated using the Mann–Whitney U test; results were considered significant at *p* < 0.05; ASA: acetylsalicylic acid; OR-1855: active metabolite of levosimendan; OR-1896: active metabolite of levosimendan; mix: blood with levosimendan and both active metabolites; ADP: adenosine diphosphate; LTA: impedance aggregometry; VASP: vasodilator-stimulated phosphoprotein.

**Table 3 ijms-25-01824-t003:** A complete list of inclusion and exclusion criteria for participation in the study.

Inclusion Criteria
Men or non-pregnant women aged 18–80 years
Provision of informed consent prior to any study-specific procedures
Diagnosis of coronary artery disease
**Exclusion criteria**
Treatment with ticlopidine, clopidogrel, prasugrel, or ticagrelor within 14 days before the study enrolment
Current treatment with oral anticoagulant or chronic therapy with low-molecular-weight heparin
Active bleeding
History of intracranial hemorrhage
Recent gastrointestinal bleeding (within 30 days)
History of coagulation disorders
Platelet count less than <100 × 10^3^/mcl
Hemoglobin concentration less than 10.0 g/dL
History of moderate or severe hepatic impairment
History of major surgery or severe trauma (within three months)
Kidney disease requiring dialysis
Respiratory failure
History of severe chronic heart failure (NYHA class III or IV)
Concomitant therapy with strong CYP3A inhibitors (ketoconazole, itraconazole, voriconazole, telithromycin, clarithromycin, nefazadone, ritonavir, saquinavir, nelfinavir, indinavir, atazanavir) or strong CYP3A inducers (rifampicin, phenytoin, carbamazepine, dexamethasone, phenobarbital) within 14 days and during study treatment

## Data Availability

The data presented in this study are available from the corresponding author on reasonable request. (Joanna Sikora; joanna.sikora@cm.umk.pl).

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
