# Peer review of "Impact of Levosimendan and Its Metabolites on Platelet Activation Mechanisms in Patients during Antiplatelet Therapy—Pilot Study"

_ijms, 2024, doi:10.3390/ijms25031824_

Round 1

Reviewer 1 Report

Comments and Suggestions for Authors

Dear authors, thanks for the opportunity to revise you paper entitled "Impact of levosimendan and its metabolites on platelet activation mechanisms in patients during antiplatelet therapy – pilot study". In this study authors have tried to understand if levosimendan and its metabolites have an effect on platelet function in CAD patients who underwent different anti platelet therapy. The idea and the argument is of great interest, but there are some major concerns that authors must address before reconsideration of their manuscript:

- English language must be extensively revised throughout the entire manuscript, necessarily with the help of a native speaker.

- There is no sample size calculation (20 patients enrolled without any explanation on number).

- Introduction is way too long. Too many in depth background that should be moved to the discussion part. 

- Methods part should be moved before results part. Is the study registered on a public registry such as clinical trials.gov?

- Authors have tried to understand if levosimendan and metabolites have effect on platelet function in patients who already underwent anti-platelet drugs. They divided patients with CAD who underwent single (ASA) or double (ASA + clopidogrel) AP therapy. But what did they expect to find? Did they wanted to compare groups? Interpreting results: Do authors suppose that levosimendan is more effective on platelet function in patient under double anti aggregation therapy? If yes, any thoughts why?

- In the discussion part authors state that: "The study of two new groups of patients, one of which used only ASA as an antiplatelet drug, and the other had dual antiplatelet therapy (ASA and clopidogrel), indicates that the antiplatelet therapy used has synergy-antiplatelet effect of levosimendan. " They cannot state that, because they did not study patients who were not under anti platelet therapy.

Minor comments:

-Introduction: Line 36 some of these are not "indications" but "area" where the drug is used to the presence of critical patients. Reformulate. Line 45: in all these cases the drug had beneficial effects due to the fact that patients had impaired cardiac function! Be aware of this. Lines 58-60: avoid to use direct questions. Lines 88 and following: express the aim in a clearer way. Move to discussion everything that is unnecessary. 

-Did the patients used other anticoagulants during the study? enoxaparin, etc..

-Conclusion is too strong. Please, reformulate in a smoother way.

- Please, add this important citation on hemodynamic monitoring during inodilator drugs administration: 10.23736/S0375-9393.23.17671-1 

Comments on the Quality of English Language

English language must be extensively revised throughout the entire manuscript, necessarily with the help of a native speaker.

Author Response

Dear Reviewer, thank you for your critical review. Below please find our detailed response:

  • English language must be extensively revised throughout the entire manuscript, necessarily with the help of a native speaker.”

We have carefully rewritten the manuscript. We had the support of a colleague who is a professional English translator.

  • “There is no sample size calculation (20 patients enrolled without any explanation on number).”

Thank you for sharing your feedback regarding the sample size determination for our pilot study. While we agree that having a sound justification for sample size is essential, it is worth noting that our pilot study is a preliminary study for a larger project. Therefore, we have determined the sample size based on feasibility goals and literature reports. According to the literature reports, the recommended number of participants for a pilot study, as suggested by Kieser and Wassmer, is between 20-40 [Kieser M, Wassmer G. Using the upper confidence limit for the variance from a pilot sample for sample size determination. Biom J 1996; 8: 941–949].

  • “Introduction is way too long. Too many in depth background that should be moved to the discussion part.”

Thank you for your opinion. Therefore, we decided to rewrite the Introduction section.

  • “Methods part should be moved before results part. Is the study registered on a public registry such as clinical trials.gov?”

We moved the Method sections to the appropriate place.

We did not register the study on clinical trials gov because the experiment is not a clinical trial (it is an in vitro study). Additionally, this is a pilot study.

  • “Authors have tried to understand if levosimendan and metabolites have effect on platelet function in patients who already underwent anti-platelet drugs. They divided patients with CAD who underwent single (ASA) or double (ASA + clopidogrel) AP therapy. But what did they expect to find? Did they wanted to compare groups? Interpreting results: Do authors suppose that levosimendan is more effective on platelet function in patient under double anti aggregation therapy? If yes, any thoughts why?”

We added an explanation to the manuscript:

We conducted the study on blood obtained from patients with CAD due to their clinical condition, which may have different platelet properties than blood from healthy people. We selected two classes of antiplatelet drugs with different mechanisms of action to verify whether any of these pathways are related to the antiplatelet effect of levosimendan. Although platelets collected from patients were inhibited by the antiplatelet therapy used (aspirin as a thromboxane inhibitor acting on the TxA2 receptor and clopidogrel as an ADP receptor agonist acting on the ADP receptor), levosimendan still additionally inhibited the action of platelets. This suggests that its platelet mechanism of action is different (through other receptors or a different mechanism of action).

“In the discussion part authors state that: "The study of two new groups of patients, one of which used only ASA as an antiplatelet drug, and the other had dual antiplatelet therapy (ASA and clopidogrel), indicates that the antiplatelet therapy used has synergy-antiplatelet effect of levosimendan. " They cannot state that, because they did not study patients who were not under anti platelet therapy.”

Thank you for this suggestion. This sentence indicates that we jumped to conclusions. We changed this part of the discussion:

“The study of two new groups of patients, one of which used only ASA as an antiplatelet drug and the other had dual antiplatelet therapy (ASA and clopidogrel) is particularly important in the case of antiplatelet therapy, where additional administration of levosimendan may significantly increase the risk of bleeding.”

  • “Introduction: Line 36 some of these are not "indications" but "area" where the drug is used to the presence of critical patients. Reformulate. Line 45: in all these cases the drug had beneficial effects due to the fact that patients had impaired cardiac function! Be aware of this. Lines 58-60: avoid to use direct questions. Lines 88 and following: express the aim in a clearer way. Move to discussion everything that is unnecessary. “

We have carefully rewritten the Introduction and suggested error was corrected:

  • “Did the patients used other anticoagulants during the study? enoxaparin, etc..”

Table 1 contains the exclusion criteria, including current treatment with oral anticoagulant or chronic therapy with low-molecular-weight heparin.

We also added to the text essential information: “Key exclusion criteria included ongoing (or terminated within preceding 14 days) treatment with any P2Y12 receptor inhibitor, treatment with oral or parenteral anticoagulants, history of intracranial hemorrhage or recent (defined as last 30 days) gastrointestinal hemorrhage, coagulation disorders at the time of screening.”

  • “Conclusion is too strong. Please, reformulate in a smoother way.”

We have rewritten the Conclusion section.

“The results of the conducted study indicate that a mechanism of levosimendan affects the function of platelets, causing higher inhibition of platelet receptors and, thus, their aggregation. It is essential to clarify whether levosimendan may affect platelets due to the need to maintain a balance between bleeding and thrombosis in patients treated with levosimendan. It is especially important in the case of perioperative bleeding. It is a fundamental observation because the concentrations of the metabolites after stopping the infusion are still increasing and reach maximum concentrations 48–78 hours after starting a 24-hour infusion. Further research is warranted to put these findings into a clinical perspective.”

  • Please, add this important citation on hemodynamic monitoring during inodilator drugs administration: 10.23736/S0375-9393.23.17671-1 “.

Thank you for your suggestion. The additional reference was added in the proper section.

Reviewer 2 Report

Comments and Suggestions for Authors

This article by Polish authors described an interesting topic, influence of levosimendan on platelet reactivity among patients receiving mono or dual antiplatelet therapy.

The manuscript is generally well written.

I have only several minor comments.

Author Response

Dear Reviewer, thank you for your critical review. Below please find our detailed response:

  • The authors did not mention the data about previous antiplatelet medication among the patients who were included in the study.”

Table 1 contains the exclusion criteria, including treatment with ticlopidine, clopidogrel, prasugrel, or ticagrelor within 14 days before the study enrolment.

We also added to the text essential information: “Key exclusion criteria included ongoing (or terminated within preceding 14 days) treatment with any P2Y12 receptor inhibitor, treatment with oral or parenteral anticoagulants, history of intracranial hemorrhage or recent (defined as last 30 days) gastrointestinal hemorrhage, coagulation disorders at the time of screening.”

  • The authors also did not precisely state the time-period between medication taking and blood sampling for laboratory analysis.”

Thank you for that suggestion, appropriate information was added to the text: “After the eligibility screening and the provision of informed consent, a blood sample for evaluations was collected 2 hours after the drug administration.”

-Why the authors decided to choose the doses of 300 mg of Aspirin and 75 mg of Clopidogrel?.”

Thank you for that question. We added appropriate information to the text: “Each study participant will be treated according to the latest ESC guidelines for managing patients presenting with CAD.”

  • “The Introduction part should be shorter.”

Thank you for your opinion. Therefore, we decided to rewrite the Introduction section.

  • The materials and methods section should be moved in the place between Introduction and Results section.

We moved the Method sections to the appropriate place.

  • “In the materials and methods section, subsection Statistical analysis, the sentence “Statistical calculations were performed using the Statistica 13 package (StatSoft, 296 Tulsa, OK, USA) and Matlab R2014 (Matlab and Statistics Toolbox Release 2014, The 297 MathWorks Inc., Natick, MA, USA)” should be move to the end of the paragraph..”

We have placed the sentence you mentioned at the end of the paragraph.

  • “In the Materials and Methods section, subsection Study design and population, the authors mentioned that patients who were included in the study were patients with coronary artery disease (CAD). But in Figure 1 they stated that patients who were included in the study were patients with CAD/patients with cardiovascular risk factors. That must be aligned. “

I appreciate you bringing Figure 4 to our attention. Please note that our study was limited to coronary artery disease (CAD) patients, and we have since corrected Figure 4 based on your suggestion.

  • “I suggest modification of the title of Table 2 in order to be clearer. Instead of “Comparison of platelet reactivity results in both groups”, I suggest “Comparison of platelet reactivity results described as p values in both groups”. I suppose that values in Table 2 are actually p values.”

Thank you for your suggestion. We have updated Table 2's title and added 'p-value' to the second and seventh rows for clarification.

Round 2

Reviewer 1 Report

Comments and Suggestions for Authors

Authors have successfully addressed all the comments raised. Thanks again for the opportunity to revise your paper.